# Active Immunization Using TRPM2 Peptide Vaccine Attenuates Atherosclerotic Progression in a Mouse Model of Atherosclerosis

**DOI:** 10.3390/vaccines13030241

**Published:** 2025-02-26

**Authors:** Fan Ying, Yunting Zhang, Xiao Li, Zhaoyue Meng, Jingxuan Li, Chun-Yin Lo, Wentao Peng, Xiaoyu Tian, Xiaoqiang Yao

**Affiliations:** 1School of Biomedical Sciences, Heart and Vascular Institute, Li Ka Shing Institute of Health Science, Faculty of Medicine, The Chinese University of Hong Kong, Hong Kong, China; yunting@link.cuhk.edu.hk (Y.Z.); 18826235169@163.com (X.L.); zhaoyuemeng@cuhk.edu.hk (Z.M.); jingxuanli@link.cuhk.edu.hk (J.L.); lcy_lo@cuhk.edu.hk (C.-Y.L.); 1155168644@link.cuhk.edu.hk (W.P.); xytian@cuhk.edu.hk (X.T.); 2Department of Laboratory Medicine, The Second Affiliated Hospital of Chongqing Medical University, No. 74 Linjing Road, Yuzhong District, Chongqing 400010, China; 3Mengniu Institute of Nutrition Science, Global R&D Innovation Center, Inner Mongolia Mengniu Dairy (Group) Co., Ltd., Hohhot 011500, China; 4School of Medicine, The Chinese University of Hong Kong, Shenzhen 518172, China

**Keywords:** TRPM2, immunization, peptide vaccine, atherosclerosis

## Abstract

**Background/Objective**: Atherosclerosis is one of the leading causes of cardiovascular diseases and mortality around the world. One exciting strategy for atherosclerosis treatment is immunotherapy, especially active immunization. Active immunization relies on the delivery of antigens in a vaccine platform to introduce humoral and cellular immunity, alleviating atherosclerotic progression. Transient receptor potential channel isoform M2 (TRPM2) is an ROS-activated Ca^2+^-permeable ion channel that can promote atherosclerosis via stimulating vascular inflammation. In the present study, we developed a strategy of active immunization with the TRPM2 E3 domain peptide in a vaccine platform, aiming to induce the endogenous production of anti-TRPM2 blocking antibody in mice in vivo, consequently inhibiting TRPM2 channel activity to alleviate atherosclerotic progression. **Methods**: ApoE knockout mice were fed with a high cholesterol diet to develop atherosclerosis. The mice were injected with or without the E3 peptide vaccines, followed by analysis of atherosclerotic lesion by en face Oil Red O staining of the whole aorta and histologic analysis of thin tissue sections from aortic roots. **Results**: The results show that immunization with a pig TRPM2 E3 region-based peptide (P1) could effectively alleviate high cholesterol diet-induced atherosclerosis in ApoE knockout mice. We worked out the best vaccine formulation for the most effective atheroprotection, namely P1 at the dose of 67.5 µg per mouse (2.5 mg/kg body weight) with aluminum salts as adjuvant. **Conclusions**: The present study provides a novel target TRPM2 for peptide vaccine-based anti-atherosclerotic strategy and lays the foundation for future preclinical/clinical trials using TRPM2 E3 P1 vaccine for a potential therapeutic option against atherosclerosis.

## 1. Introduction

Atherosclerosis is one of the leading causes of cardiovascular diseases and mortality around the world [1]. In atherosclerosis, fatty and fibrous matter accumulates to form plaques that impede the blood flow and provoke thrombotic events, eventually leading to myocardial infarctions and strokes [2,3]. Hypercholesterolemia and excessive oxidative stress in arterial walls are among the main causative factors for atherosclerosis [4,5]. During atherosclerotic development, plasma low density lipoprotein (LDL) is oxidatively modified into oxidized LDL (oxLDL) in the subendothelial space, where it attracts leukocytes, including monocytes, to vascular walls [3,5]. The monocytes differentiate into macrophages to produce excessive reactive oxygen species (ROS), consequently stimulating multiple pathological events, such as inflammation, vascular cell death, vascular wall hyperplasia and narrowing/occlusion of blood vessel lumen [5].

Atherosclerosis is often treated with cholesterol-lowering pharmacological agents such as statins and anti-platelet medications such as clopidogrel (Plavix) [6,7]. Recently, inclisiran, which is a novel siRNA drug that targets proprotein convertase subtilisin/kexin type 9 gene expression, has also been used in clinical practice to reduce blood cholesterol levels [8,9]. However, although these agents can slow down the progression of atherosclerosis to a certain degree, they cannot cure the disease. The development of new therapeutic strategies is being attempted.

One exciting strategy for atherosclerosis treatment is immunotherapy, especially active immunization [10,11,12,13,14]. In this regard, researchers have been developing atherosclerosis-related vaccines for the past two decades [13,14]. Most of these studies target LDL-related antigens, such as oxLDL, apolipoprotein B-100 and proprotein convertase subtilisin/kexin type-9 serine protease (PCSK9) [10,11,12,13,14,15]. However, to date, most of these vaccine-based studies for atherosclerosis are still in preclinical stages using animal models. None of these vaccines have been approved for clinical usage. The idea of developing vaccination strategies to treat atherosclerotic cardiovascular disease is still in its infancy.

Transient receptor potential channel isoform M2 (TRPM2) is a ROS-sensitive Ca^2+^-permeable cation channel which is activated by ADP-ribose and an endogenous ROS H_2_O_2_ [16,17]. A substantial amount of evidence from us and others indicates that the activity of TRPM2 contributes to vascular inflammation and vascular cell death [17,18]. Indeed, it has been shown that TRPM2 stimulates the production of inflammatory chemokines [19], enhances neutrophil migration across the endothelial barrier [20] and promotes vascular cell death via triggering cellular Ca^2+^ overload [18]. More recently, with the use of TRPM2 knockout mice, we and others have demonstrated that TRPM2 promotes hypercholesterolemia-induced atherosclerosis via stimulating vascular inflammation [21,22].

TRP channels and many voltage-gated potassium channels share a typical secondary structure of six transmembrane domains. The E3 domain is a region that is hydrophilic, a C-terminal to the fifth transmembrane domain (S5) and close to the predicted ion pore [23]. We have previously immunized rabbits with an E3 region-based rat TRPM2 short peptide to generate a polyclonal anti-TRPM2 antibody [18,24]. The generated rabbit antibody TM2E3 can effectively inhibit the activity of human and rat TRPM2 channels in patch clamp recording and Ca^2+^ measurement study in vitro [16 Sun L]. Intriguingly, TM2E3 also inhibits several cellular processes that are closely associated with atherosclerotic development, including the proliferation and migration of vascular smooth muscle cells and ROS-induced vascular cell death in murine models [24].

In the present study, we made the first attempt to develop the strategy of active immunization with the TRPM2 E3 peptide in a vaccine platform, aiming to induce endogenous production of the anti-TRPM2 blocking antibody in mice in vivo. ApoE knockout mice were fed with a high cholesterol diet to establish a mouse atherosclerosis model. The mice were injected with the E3 peptide vaccines in order to determine whether active immunization with the E3 peptides could attenuate atherosclerosis. The schematic experimental schedule is displayed as Figure 1a. The results from the experiments demonstrate that active immunization with a pig TRPM2 E3 region-based peptide P1 could indeed reduce the atherosclerotic progression in a mouse model.

## 2. Materials and Methods

### 2.1. Animals and Immunization

ApoE knockout mice with a C57/BL background were obtained from The Jackson Laboratory (Bar Harbor, ME, USA) and transported to the Laboratory Animal Services Centre at the Chinese University of Hong Kong. To induce atherosclerosis, the mice were fed either a normal chow diet or a high-cholesterol diet (catalog no. D12336, Research Diets, New Brunswick, NJ, USA) starting at eight weeks of age. The mice were kept on a 12 h light–dark cycle with ad libitum access to food and fresh water. All animal experiments were approved by the Hong Kong Department of Health and the Animal Experimentation Ethics Committee of the Chinese University of Hong Kong.

Peptide antigens with a purity of >99% were commercially synthesized. The sequence was validated by Mass Spec. The peptides were conjugated to keyhole limpet hemocyanin (KLH) (GL Biochem Ltd., Shanghai, China). The KLH-conjugated peptides were mixed with an equal volume of Complete Freund’s adjuvant or with aluminum hydroxide adjuvant, then immunized to ApoE knockout mice subcutaneously at a dose of 67.5 µg per mouse (2.5 mg/kg body weight) or 135 µg per mouse (5 mg/kg body weight) (day 0). Two boost doses with the same concentration of peptide antigens were injected into the same mice at day 21 and day 42, respectively. For the mice using Complete Freund’s adjuvant in the first injection, the adjuvant for two boost doses of injection was Incomplete Freund’s adjuvant with the peptide/adjuvant volume ratio of 1:1. For the mice using aluminum hydroxide adjuvant in the first injection, the adjuvant for two boost doses of injection was aluminum hydroxide adjuvant with the peptide/adjuvant volume ratio of 1:1. The animals were fed a high-cholesterol diet for an additional three months to promote atherosclerotic development, after which they were sacrificed for sample analysis (Figure 1a).

### 2.2. En Face Oil Red O Staining of Whole Aorta

ApoE knockout mice were euthanized by CO_2_ asphyxiation. The aortas were carefully dissected in cold PBS and opened to expose the atherosclerotic plaques. After fixation in 4% formaldehyde for 10 min at 4 °C, the tissues were first rinsed in distilled water for 10 min, followed by a rinse in 60% isopropanol. The aortas were then stained with Oil Red O for 20 min with gentle shaking, rinsed in 60% isopropanol, and washed three times in water. The samples were placed on glass slides with the endothelial surface facing up. Images were captured using a SONY RX100VI Camera (SONY, Hong Kong, China). Plaque areas were analyzed using National Institutes of Health ImageJ 1.52a software (ImageJ, Bethesda, MD, USA), and the plaque area was expressed as a percentage of the total vascular area.

### 2.3. Measurements of Serum Lipids

Blood was collected from the celiac vein, and serum was separated by centrifugation at 2000× *g* for 10 min at room temperature. The serum lipid profile, including total cholesterol, LDL-cholesterol and triglycerides, was determined using commercially available assay kits (Stanbio, Boerne, TX, USA).

### 2.4. Elisa-Based Quantification of Antibody Production in Mice

A 96-well microtiter plate was coated overnight at 4 °C with synthetic peptides based on the TRPM2 E3 region in carbonate buffer. After thorough washing, the plate was blocked with BSA in PBST at 37 °C for 1 h and then dried for later use. Polyclonal antiserum from mice immunized with different peptide vaccines or control KLH was serially diluted and added to the plate, followed by overnight incubation at 4 °C. After washing the plate three times, HRP-conjugated goat anti-mouse secondary antibody was added and incubated at 37 °C for 1 h. The optical density of the plate was measured using a time-resolved plate reader.

### 2.5. Generation of a Polyclonal Antibody TM2E3-P1

A polyclonal antibody TM2E3-P1 was raised in rabbits through a commercial source (General Biol, Anhui, China) replicating a previously described strategy [23]. Briefly, a peptide corresponding to a pig TRPM2 E2 region (CHNERRVEWIFRGAVYQ), named P1, was synthesized and conjugated to keyhole limpet hemocyanin at General Biol, Anhui, China. The conjugated P1 peptide was injected subcutaneously into the back of a rabbit, followed by two booster doses. TM2E3-P1 antiserum was collected 4 weeks after the second booster. The polyclonal antibody TM2E3-P1 was purified from the TM2E3-P1 antiserum, while pre-immune IgG was purified from pre-immune serum using a protein G column.

### 2.6. Cytosolic Ca^2+^ Measurement

Mouse microvascular endothelial cell line H5V was cultured in Dulbecco’s modified Eagle medium (DMEM) (Gibco, Invitrogen, Waltham, MA, USA) supplemented with 10% fetal bovine serum (FBS) (Invitrogen, MA, USA) and penicillin-streptomycin (Invitrogen, MA, USA). Human umbilical vein endothelial cells (HUVECs) were cultured in endothelial cell growth medium supplemented with 1% bovine brain extract. The cells were incubated in the dark with 10 µM Fluo-4 (Invitrogen, MA, USA) and 0.02% Pluronic F-127 in Ca^2+^-free physiological saline solution (Ca^2+^-free PSS) at 37 °C for 30 min. After incubation, the cells were placed in Ca^2+^-free PSS, challenged with 500 μM H_2_O_2_ to induce intracellular Ca^2+^ release, and then treated with 2 mM Ca^2+^ for extracellular Ca^2+^ entry. The Ca^2+^-free PSS contained the following (in mM): 140 NaCl, 5 KCl, 1 MgCl_2_, 10 glucose, 0.2 EGTA, 5 HEPES and pH 7.4. Fluorescence intensity and the relative signal (F1/F0 ratio) were measured using an Olympus FV1000 confocal microscope.

### 2.7. Histologic Analysis

The proximal aorta attached to the heart was harvested and frozen in optimal cutting temperature compound. Serial 8 µm sections were stained with Oil Red O to detect lipid droplets and Masson’s Trichrome to assess collagen deposition. For immunostaining of the frozen sections, CD68 (1:300, Proteintech, 28058-1-AP), MPO (1:100, Proteintech, 22225-1-AP) and PCNA (1:100, Proteintech, 10205-2-AP) antibodies, along with secondary antibodies, were applied. Images from five random visual fields were captured under a microscope. Quantification of lesion area and necrotic core size was performed using ImageJ software (National Institutes of Health, Bethesda, MD, USA).

### 2.8. Statistical Analysis

A normality test was performed before comparing two groups using the D’Agostino and Pearson omnibus normality test (Prism software 8.3 recommended) or Shapiro–Wilk test or Kolmogorov–Smirnov test. For normally distributed data, an unpaired two-tailed Student’s *t*-test was used. If the data did not meet normality, non-parametric tests were applied. Differences among three or more groups were analyzed by one-way analysis of variance (ANOVA) followed by Dunnett’s multiple comparisons test. Statistical analyses were conducted using GraphPad Prism 8.3 (GraphPad Software, San Diego, CA, USA). *p* values < 0.05 were considered statistically significant. For en face Oil Red O and histological analyses (Figures 2, 5 and 6), five to ten mice were used per treatment group, with data presented as mean ± standard deviation (SD). For Ca^2+^ analysis, antibody titration and inflammatory cytokine analysis (Figure 3, Appendix A), data are represented as mean ± standard error of the mean (SEM).

## 3. Results

### 3.1. Design of the Peptide Antigens

We designed several E3 region-based peptide vaccines. The designed peptides were based on the TRPM2 E3 domain of different animal species (pig or rabbit or human) and from different regions of the E3 domain. For animal species consideration, rodent TRPM2 peptide was avoided because these peptides might not be able to elicit immune responses in the mice themselves. In consideration of different E3 domains, the 40 amino acid-long E3 domain was divided into two regions. Region 1 is located near the N-terminal end of the E3 region, whereas region 2 is close to the C-terminal end of the E3 peptide (Figure 1b). The length of peptides was 16–21 amino acids, similar to our previous reports [18,24]. Here, P1 stands for pig region 1 peptide, H2 stands for human region 2 peptide, whereas R1 and R2 represent rabbit region 1 and rabbit region 2, respectively. A comparison of the TRPM2 amino sequence shows that the pig peptide P1 sequence is highly similar to the corresponding sequences of mouse TRPM2 with 75% amino acid, which is identical, whereas rabbit peptide sequences (R1 and R2) are relatively distant from the corresponding sequences of mouse TRPM2, with 44% and 37% amino acid identity, respectively.


### 3.2. Immunization with Different Peptide Vaccines to Suppress Atherosclerotic Progression Based on en Face Oil Red O Staining of Whole Aortas

We immunized the TRPM2 E3 region-based peptide vaccines, including P1, R1, R1+R2 and H2, in ApoE knockout mice subcutaneously at the dosage of 135 µg per mouse (5 mg/kg body weight). The effect of peptide immunization on the development of atherosclerotic plaques was examined in isolated aortas of animals using en face Oil Red O staining. The results show that, compared with the control with no peptide immunization, immunization with P1 significantly suppressed the atherosclerotic progression, with the atherosclerotic lesion area reduced from 38% to 28%, a relative reduction of (38–28%)/38%, which is ~26% (Figure 2a), whereas immunization with other peptides did not significantly reduce the atherosclerotic progression in en face Oil Red O staining (Figure 2a).

We also measured the blood lipid profile of ApoE knockout mice treated with different vaccine peptides. Compared with the mice fed with a normal chow diet, high cholesterol diet feeding indeed elevated the serum cholesterol and low-density lipoprotein (LDL) cholesterol level (Figure 2b,c), but had no effect on the serum triglyceride level (Figure 2d). There were no differences in serum lipid profiles among different vaccine treatment groups (Figure 2b–d).


### 3.3. Polyclonal Antibody TM2E3-P1 Inhibited the H_2_O_2_-Induced Ca^2+^ Entry in Endothelial Cells

Because the pig TRPM2 E3 region-based peptide P1 vaccine is effective in suppressing atherosclerotic progression, we next focused on the P1 vaccine. We measured the production of IgG antibodies in mice after P1 vaccine immunization. Results show that the P1 vaccine elicited modest antibody production compared with KLH control, with the antibody titer of 1:620 (Figure 3a).

We then used the P1 peptide as an antigen and immunized it to rabbit to generate the E3 region-based polyclonal anti-TRPM2 antibody, named TM2E3-P1. The serum was collected and the antibody was purified from the rabbits. We determined whether TM2E3-P1 could inhibit the activity of TRPM2 channels in mouse endothelial cells (H5V) and human umbilical vein endothelial cells (HUVECs). H_2_O_2_ was used to activate TRPM2-mediated Ca^2+^ entry in endothelial cells, which was validated previously by us [18,21]. A two-step protocol was used to differentiate H_2_O_2_-stimulated extracellular Ca^2+^ entry from H_2_O_2_-stimulated Ca^2+^ released from intracellular Ca^2+^ stores. The cells were first bathed in a Ca^2+^-free physiological saline solution (0Ca^2+^-PSS). The application of H_2_O_2_ at 500 µM initiated a cytosolic Ca^2+^ rise, which presumably was due to H_2_O_2_-stimulated Ca^2+^ store release (Figure 3b,d). Then, 2 mM Ca^2+^ was added back to the extracellular bath, causing the second cytosolic Ca^2+^ rise, which was due to Ca^2+^ entry (Figure 3b,d). As a control, some cells were bathed in a Ca^2+^-free solution without H_2_O_2_ pretreatment; the addition of extracellular Ca^2+^ to these cells only induced a small cytosolic Ca^2+^ rise, much smaller than the cells with H_2_O_2_ pretreatment (Figure 3b–e). Therefore, the second Ca^2+^ rise in response to extracellular Ca^2+^ addback in H_2_O_2_-pretreated cells mostly represented the H_2_O_2_-stimulated Ca^2+^ entry. Importantly, our results showed that, when compared with the controls that were pretreated with rabbit preimmune IgG, TM2E3-P1 pretreatment for 15 min at the concentration of 10 µg/mL substantially reduced the H_2_O_2_-stimulated Ca^2+^ entry in both H5V cells (Figure 3b,c) and HUVECs (Figure 3d,e). These data confirmed that the TM2E3-P1 antibody could inhibit the activity of mouse and human TRPM2 channels. Note that the peptide sequence of P1 is very similar to the corresponding sequence of mice (75% identical, Figure 1b) and humans (81% identical). It is not surprising that TM2E3-P1 was able to inhibit mouse and human TRPM2.

We also measured the production of IgG antibodies in mice after immunization with R1, R1+R2 and H2 vaccines. R1 did not effectively elicit antibody production. R2 and R1+R2 peptide vaccines generated high amounts of antibodies with titers of 1: >20,000 (Appendix A). H2 elicited a massive amount of antibody production with an antibody titer of 1: >40,000 (Appendix A). Coincidently, we also observed that 7 out of 20 animals immunized with H2 peptide vaccine died. Reports show that an overactive immune response can sometimes trigger a cytokine storm, causing excessive systemic inflammation and tissue damage [25]. Therefore, we examined whether H2 peptide vaccine could elicit excessive production of inflammatory cytokines including TNF-α, IL-1β and IL-6. The results showed that H2 peptide vaccines caused a certain increase in the serum level of TNF-α, but not in serum levels of IL-1β and IL-6 (Appendix A). Because the effect of the H2 peptide vaccine on the production of inflammatory cytokines was not very consistent, it is unlikely that excessive immune response was the main reason for animal death. Further studies are needed to find the underlying reason.


### 3.4. The P1 Vaccine Reduced Atherosclerotic Plaques in Aortic Roots

To further validate the anti-atherosclerotic effect of the P1 peptide vaccine in high cholesterol-fed ApoE knockout mice, frozen thin sections were prepared from aortic roots and then subjected to Oil Red O staining for plaque area analysis and Masson’s trichrome staining for fibrosis and necrotic core and fibrosis evaluation. The results demonstrate that the P1 vaccine immunization reduced the atherosclerotic plaque area (Figure 4a) and the necrotic core size (Figure 4b) at the aortic roots.

### 3.5. Comparison of Different P1 Vaccine Doses and Adjuvants in Suppressing Atherosclerotic Progression

Two different dosages of P1 peptide vaccines were tested, 67.5 µg per mouse (2.5 mg/kg body weight) and 135 µg per mouse (5 mg/kg body weight). Meanwhile, two commonly used adjuvants, aluminum hydroxide adjuvant and Freund’s adjuvant, were also evaluated. Aluminum hydroxide adjuvant is among the most widely applied adjuvants in human vaccine design, while Freund’s adjuvant is the most commonly used adjuvant in animal immunization.

Results from the en face Oil Red O staining of whole aortas showed that immunization with P1 vaccine reduced the atherosclerotic progression at the dose of 67.5 µg per mouse with either aluminum hydroxide adjuvant (Figure 5a, a net reduction by ~11%) or with Freund’s adjuvant (Figure 5c, a net reduction by ~17%). Similarly, at the peptide dose of 135 µg per mouse, the P1 vaccine also reduced on atherosclerotic development with both aluminum hydroxide adjuvant (Figure 5b, a net reduction of ~3%) and Freund’s adjuvant (Figure 5d, a net reduction of ~7%). Therefore, P1 vaccine peptide at 67.5 µg per mouse was at least equivalent or even better than 135 µg per mouse for both aluminum hydroxide and Freund’s adjuvants.

Next, we used thin tissue sections of aortic root to further investigate the effect of P1 immunization on other atherosclerotic indexes. Oil red staining of aortic root sections indicated that immunization with peptide P1 at both 67.5 µg and 135 µg per mouse reduced the atherosclerotic lesion area with comparable effect (Figure 6a). Consistently, P1 also decreased the necrotic core size with either aluminum hydroxide or Freund’s adjuvants in Masson’s trichrome, except for the combination of 67.5 µg P1 with aluminum hydroxide adjuvant (Figure 5b). Finally, we also examined the effect of P1 immunization on the expression of several atherosclerotic markers, including CD68, myeloperoxidase (MPO) and proliferating cell nuclear antigen (PNCA). CD68 and MPO could detect infiltrated macrophages and neutrophiles, respectively, whereas PCNA is a proliferating cell marker. The results show that P1 immunization reduced the expression of CD68, MPO and PCNA in the lesion regions with both aluminum hydroxide and Freund’s adjuvants (Figure 5c).


## 4. Discussion

Atherosclerosis is an immune-mediated inflammation disease involving the participation of both the humoral and cellular immune system [10,11,12,13,14]. It is tempting to consider specific strategies to modulate the immune responses to attenuate atherosclerotic progression. Previously, researchers have extensively studied the vaccine strategy that targets LDL-related antigens, such as oxLDL, apolipoprotein B-100 and proprotein convertase subtilisin/kexin type-9 serine protease (PCSK9) [10,11,12,13,14,15]. These vaccines can successfully attenuate atherosclerosis in a mouse model [10,11,12,13,14]. More recently, several groups have begun to explore the vaccines that target other important molecules/processes in atherosclerotic development, such as ANGPTL3 [26] and COL6A6 [27]. TRPM2 is expressed in monocytes, macrophages and vascular cells in vascular walls [17,18,24]. Recent studies from us and others showed that TRPM2 has a proinflammatory role in vascular walls, consequently promoting atherosclerotic progression [21,22]. Therefore, in this study, we chose to use TRPM2 peptides in a vaccine platform to activate anti-TRPM2 immune response as a means to protect against atherosclerosis.

The most important consideration in vaccine peptide design is to find the best peptide sequence. Our vaccine peptides are based on the E3 domain of TRPM2. Previously, we used the rat TRPM2 E3 domain to generate rabbit polyclonal antibody TM2E3 that can effectively inhibit the activity of TRPM2 channels [18,24]. The length of peptides was 16–21 amino acids, which is suitable for class II MHC presentation, helper T-cell activation and antibody production [28]. We expect the E3 domain-based vaccine peptide to generate antibodies in mice, inhibiting the TRPM2 channel activity in mice, and consequently exerting an anti-atherosclerotic action. Indeed, our results showed that the injection of a pig TRPM2 E3 region-based peptide P1 in a vaccine platform could effectively reduce atherosclerotic progression in high-cholesterol-fed ApoE knockout mice, based on en face Oil Red O staining of the whole aorta. Further analysis showed that the P1 vaccine reduced the size of the necrotic core and decreased the Oil Ted-positive area in tissue sections of aortic roots. Note that among all analysis methods, en face Oil Ted staining is most reliable as it represents unbiased overall atherosclerotic progression, whereas other methods rely on thin tissue section staining, which is liable to section sampling variations. Taken together, our study, for the first time, showed that the E3 domain-based TRPM2 peptide P1 can be used in a vaccine platform to attenuate atherosclerotic progression in a mouse model of atherosclerosis.

Regarding the design of vaccine peptide sequences, the E3 domain of TRPM2 is about 40 amino acid long, which we arbitrarily divided into two regions, one close to the N-terminal end of the E3 and the other close to the C-terminal end. Our results showed that two region 2-based peptide vaccines, R2 and H2, generated a good amount of antibody, but failed to significantly inhibit atherosclerotic progression. On the other hand, a region 1-based peptide vaccine P1 could effectively inhibit atherosclerotic progression. Therefore, region 1 is the better option for peptide vaccine design. We speculate that this may be related to the molecular structure of the TRP channel, in which region 1 is predicted to be exposed to the cell surface for easy antibody access, whereas region 2 is predicted to be invaginated into the pore region, where antibodies may have difficulty gaining access. Another important consideration in vaccine peptide design is the sequence similarity of vaccine peptides to that of the TRPM2 E3 region. Two competing issues need to be considered, namely, (1) immunogenicity or capability of antibody production and (2) ability of the generating antibody to recognize and bind to mouse TRPM2 for its inhibitory action. A high sequence similarity of the vaccine peptide to that of mouse TRPM2 E3 is expected to reduce immunogenicity in mice. However, the generated neutralizing antibody should be easier to recognize and bind to the mouse TRPM2 proteins for its inhibitory action. In contrast, a low sequence similarity in TRPM2 vaccine peptides to that of mouse TRPM2 should increase the immunogenicity in mice. But the generated neutralizing antibody might not be able to effectively recognize and bind to the mouse TRPM2 proteins. This could also partly explain why P1, which has very high sequence similarity to the mouse TRPM2 E3 region, could only generate a moderate amount of antibodies yet achieved an effective anti-atherosclerotic effect. In contrast, R2 and H2, which had low similarity to the mouse TRPM2 E3 region, could generate a good amount of antibody, but failed to significantly inhibit atherosclerotic progression.

Other important considerations in the vaccine regimen include dosages of vaccine peptides and adjuvants. Regarding dosage, we found that P1 at a low dose of 67.5 µg per mouse (2.5 mg/kg body weight) had an equivalent or even better effect than 135 µg per mouse (5 mg/kg body weight). Therefore, a low dose is a better choice for the TRPM2 E3 peptide vaccine, as it may reduce the possibility of unwanted side effects. For adjuvants, a comparison of two different adjuvants showed that Freund’s adjuvant appears to be slightly better than the aluminum hydroxide adjuvant (Figure 5a vs. Figure 5c; Figure 5b vs. Figure 5d). However, note that the aluminum hydroxide adjuvant has no human toxicity, and thus has been widely used in human vaccination, whereas Freund’s adjuvant may be too toxic for human applications [29]. Because our ultimate goal is to develop a human TRPM2-based vaccine, aluminum hydroxide adjuvant is the more appropriate choice.

As for the molecular mechanism of how E3 region-based peptide vaccines can inhibit atherosclerotic progression, the most likely underlying reason may involve the vaccine-induced production of the E3-targeting neutralizing antibody, which binds to TRPM2 and inhibit its activity, consequently attenuating atherosclerotic progression. Indeed, neutralizing antibody production was detected in the mouse serum after P1 vaccine. The circulating neutralizing antibody is expected to bind to endothelial TRPM2 and monocyte TRPM2 to exert its effect. Indeed, we confirmed that a rabbit polyclonal antibody TM2E3-P1 is capable of inhibiting the TRPM2-mediated Ca^2+^ influx elicited by H_2_O_2_ in mouse and human endothelial cells. These data agree with our hypothesis of the vaccine-induced neutralizing antibody being the main reason for the anti-atherosclerotic action of the P1 peptide vaccine. Alternatively, we cannot exclude the possible involvement of cellular immunity in TRPM2 E3 peptide vaccine-induced atheroprotection. Others have reported that the atheroprotection of LDL-related peptide vaccines (such as the p210 vaccine) is mostly due to cellular immunity, but not so much due to humoral immunity [10,11,12,13,14]. Further study is needed to determine whether cellular immunity is also involved in TRPM2 E3 vaccine-induced atheroprotection.

The present study represents the first attempt to use the TRPM2 E3 peptide in a vaccine platform for the purpose of suppressing atherosclerotic progression in a mouse model. Unlike most previous vaccine-related studies, which target LDL-related antigens [10,11,12,13,14,15], we targeted TRPM2, which is expressed in multiple atherosclerosis-related cell types (monocytes, macrophages, vascular endothelial cells and vascular smooth muscle cells) and is involved in ROS-related vascular inflammation [17,18,19,20,21,22]. Therefore, our TRPM2 E3 region-based vaccine targets different steps/processes in atherosclerosis compared with most other vaccines that target LDL-antigens, providing an alternative therapeutic option. However, note that among the six TRPM2 E3 region-based peptide vaccines we used, only one peptide vaccine P1 can effectively suppress the atherosclerotic progression with statistical significance, whereas another one, the H2 peptide vaccine, may have some safety concerns. Therefore, for the further development of TRPM2 E3 domain-based peptide vaccines, it is important to carefully select the specific region/amino acid sequence within the TRPM2 E3 domain to increase the effectiveness and to avoid safety concerns. Another limitation is that only a relatively small number of mice were used in the present study. Further preclinical studies are needed to test the P1 peptide vaccine in a large number of mice to determine the overall effectiveness of this vaccine in suppressing atherosclerosis and to examine any possible side effect.

## 5. Conclusions

Our present study developed a novel strategy of active immunization with TRPM2 E3 peptide in a vaccine platform to attenuate atherosclerotic progression. We worked out the best vaccine formulation for the most effective atheroprotection, namely P1 at the dose of 67.5 µg per mouse (2.5 mg/kg body weight) with aluminum hydroxide adjuvant salts as the adjuvant. Our current study lays the foundation for future clinical trials using TRPM2 peptide vaccine as a potential treatment for atherosclerosis.

## 6. Patents

Yao, X., Cheung, W.T., Ying, F., Zhang, Y., Meng, Z. (2021): TRPM2-based peptide vaccine against atherosclerosis in ApoE knockout mouse model. Chinese patent: 202110245910.6.

## Figures and Tables

**Figure 1 vaccines-13-00241-f001:**
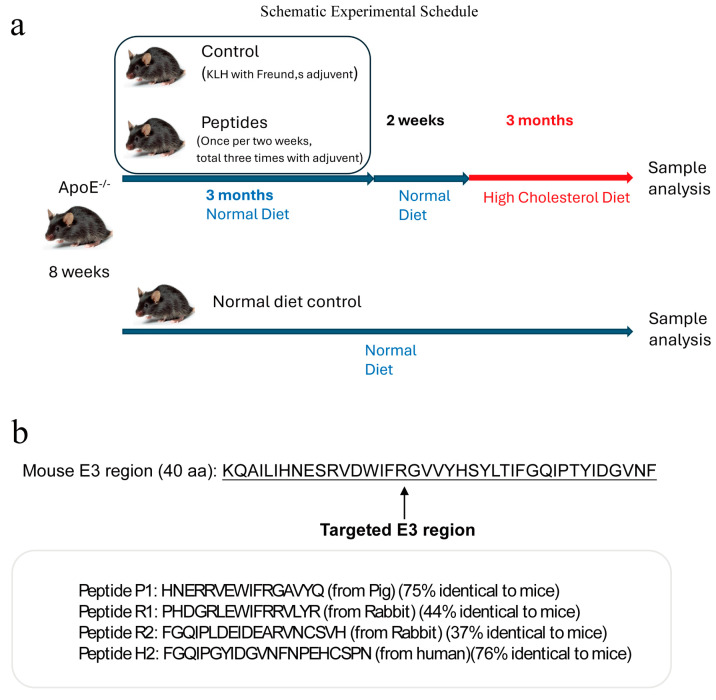
Experimental schedule (**a**) and the amino acid sequences of peptide antigens (**b**).

**Figure 2 vaccines-13-00241-f002:**
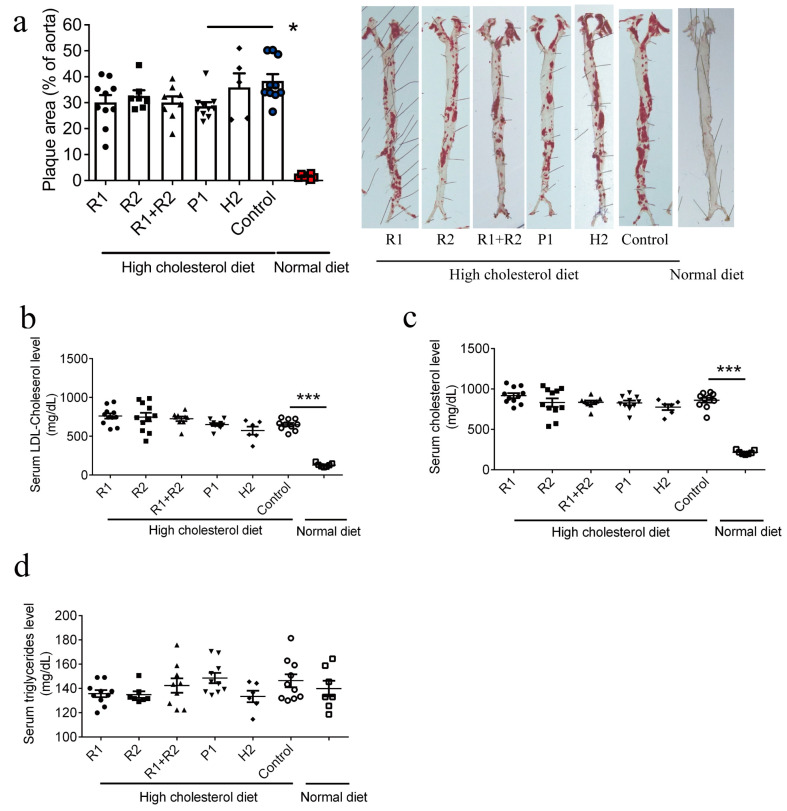
A pig TRPM2 E3 region-based peptide vaccine P1 ameliorates the development of atherosclerotic plaques in mouse whole aortas. (**a**) Quantification of atherosclerotic plaques (**left**) and representative images (**right**) of Oil Red O staining of whole aortas. (**b**–**d**) The effect of high cholesterol diet feeding and E3 region-based peptide vaccines on serum LDL cholesterol level (**b**), serum cholesterol level (**c**) and triglyceride level (**d**). Data are shown as mean ± SD (n = 5–11), * *p* < 0.05; *** *p* < 0.001.

**Figure 3 vaccines-13-00241-f003:**
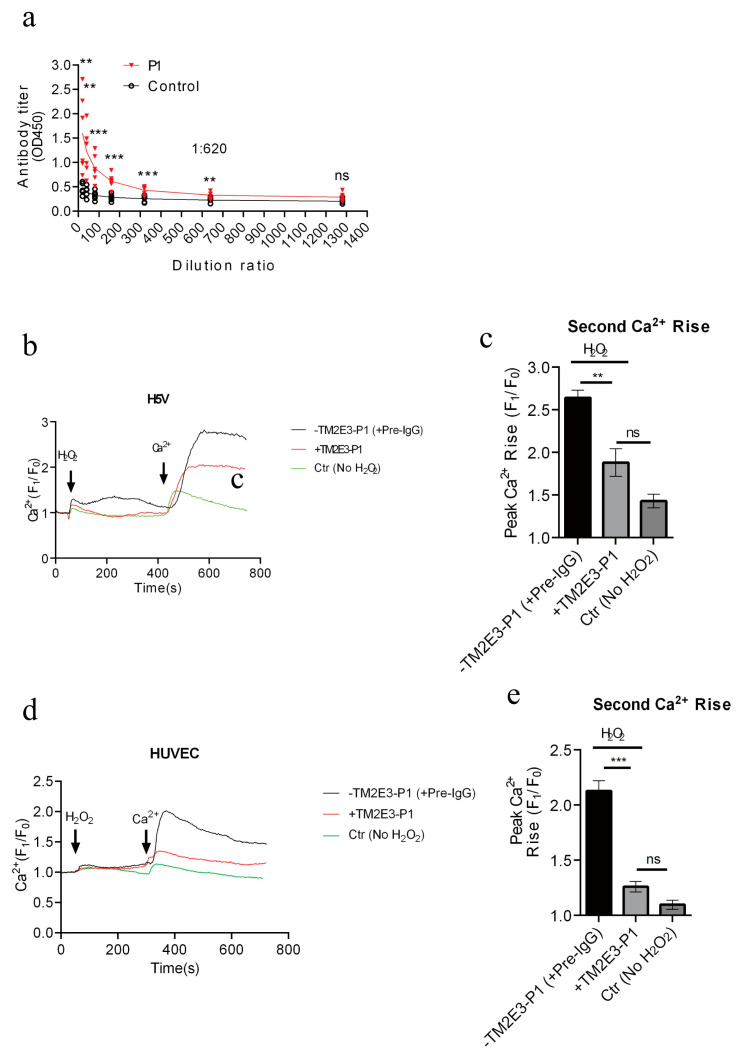
Titration of TRPM2 E3-based polyclonal antibody in mice and effectiveness of TM2E3-P1 in suppressing H_2_O_2_-stimulated Ca^2+^ entry in endothelial cells. (**a**) Titration of polyclonal antibodies produced in mice in response to P1 peptide vaccine. The mice were immunized with P1 peptide vaccine or KLH control. The polyclonal antiserum was taken from the mice, followed by Elisa-based titration vs. synthetic P1 peptide. Shown are titration curves of the antiserum at different dilutions. (**b**–**e**) Polyclonal antibody TM2E3-P1 (compared with pre-immune IgG) could effectively inhibit the H_2_O_2_-stimulated Ca^2+^ entry in mouse H5V endothelial cells (**b**,**c**) and human HUVEC endothelial cells (**d**,**e**). The cells were bathed in 0Ca^2+^-PSS and challenged by 500 µM H_2_O_2_, which elicited the first cytosolic Ca^2+^ rise due to intracellular Ca^2+^ release. Then, 2 mM Ca^2+^ was added back to initiate the second Ca^2+^ rise, which was due to Ca^2+^ entry. Shown were representative time course (**b**,**d**) and data summary for Ca^2+^ entry (**c**,**e**). Controls had no H_2_O_2_ treatment. Mean ± SEM (n = 3–7); ns, not significant; **, *p* < 0.01; ***, *p* < 0.001.

**Figure 4 vaccines-13-00241-f004:**
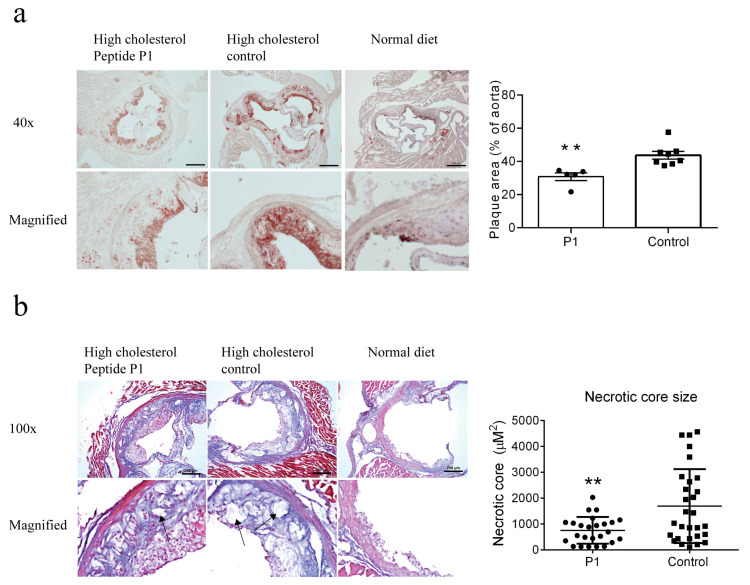
Immunization with P1 peptide vaccine reduced the development of atherosclerotic plaques in thin tissue sections of aortic roots. (**a**) Representative tissue section images (**left**) and data summary (**right**) of aortic roots stained with Oil Red O. The atherosclerotic lesion area is visualized as red. (**b**) Representative tissue section images (**left**) and data summary for necrotic sore size (**right**) of aortic roots stained with Masson’s trichrome. The collagen is stained in blue. The size of the necrotic core is quantified using Image J. Quantification was performed in five random visual fields of multiple slides prepared from five to eight mice. Scale bars are as indicated. Data are shown as Mean ± SEM (n = 5–8 in (**a**), n = 25–29 in (**b**)), ** *p* < 0.01.

**Figure 5 vaccines-13-00241-f005:**
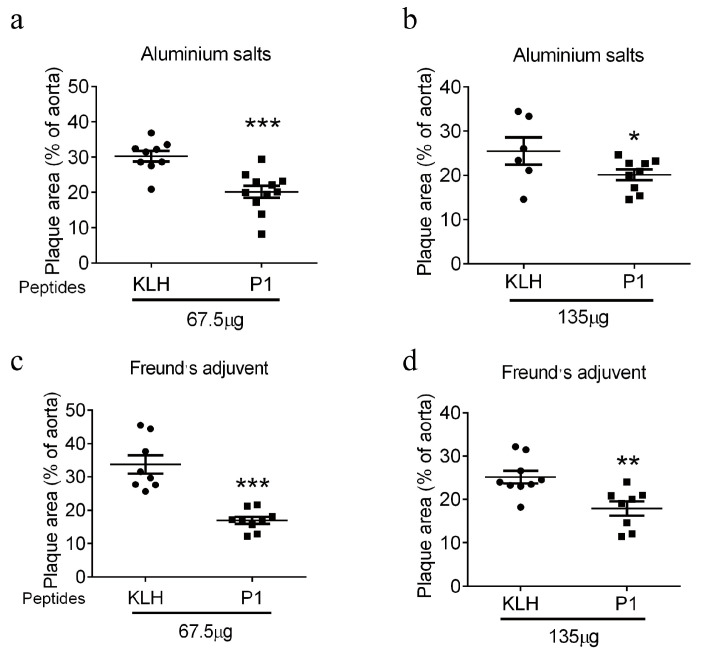
Comparison of different P1 vaccine doses and adjuvants in suppressing atherosclerotic progression based on en face Oil Red O staining of whole aortas. Two different dosages of P1 peptide vaccines were tested, 67.5 µg per mouse (2.5 mg/kg body weight) (**a**,**c**) and 135 µg per mouse (5 mg/kg body weight) (**b**,**d**). Two different adjuvants, aluminum hydroxide adjuvant (**a**,**b**) and Freund’s adjuvant (**c**,**d**), were also evaluated. Shown is the quantification of atherosclerotic plaques based on en face Oil Red O staining of whole aortas. The plaque area is quantified by using Image J. Data are shown as mean ± SD (n = 6–10), * *p* < 0.05; ** *p* < 0.01; ***, *p* < 0.001.

**Figure 6 vaccines-13-00241-f006:**
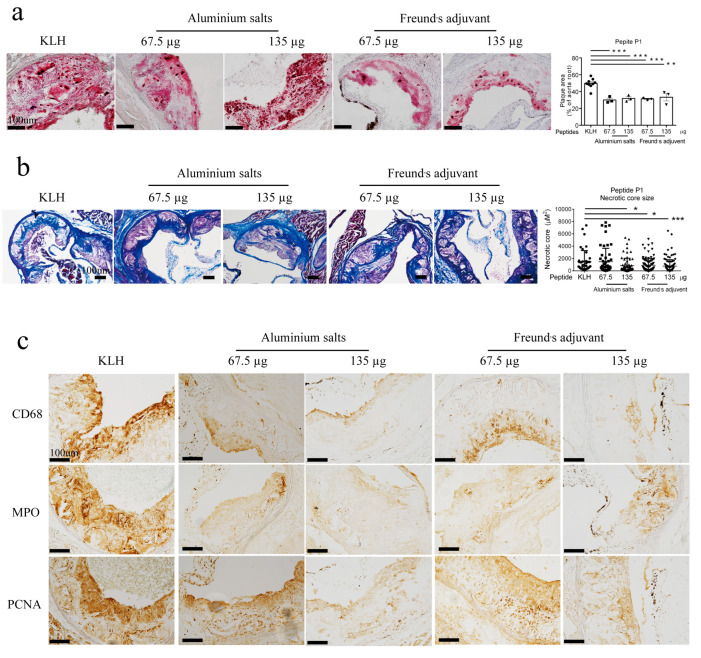
Comparison of different P1 vaccine doses and adjuvants in suppressing atherosclerotic progression based on the staining on thin tissue sections of aortic roots. (**a**) Representative images (**left**) and data summary of Oil Red O staining of thin tissue sections of mouse aortic roots. (**b**) Representative images (**left**) and data summary of Masson’s trichrome staining of thin tissue sections of mouse aortic roots. (**c**) Representative images (**left**) and data summary of immunohistochemical stains of CD68, MPO and PCNA on thin tissue sections of mouse aortic roots. The mice were treated with two different P1 vaccine doses (67.5 µg or 135 µg per mouse) and two different adjuvants (aluminum hydroxide adjuvant or Freund’s adjuvant). Quantification was performed in five random visual fields of multiple slides prepared from three to ten mice. Scale bar, 100 µm as indicated. Mean ± SD (n = 3–10 in (**a**), n = 40–120 in (**b**)), * *p* < 0.05; ** *p* < 0.01; *** *p* < 0.001.

## Data Availability

The data presented in this study are available on request from the corresponding author.

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
