# Peer review of "Active Immunization Using TRPM2 Peptide Vaccine Attenuates Atherosclerotic Progression in a Mouse Model of Atherosclerosis"

_vaccines, 2025, doi:10.3390/vaccines13030241_

Round 1

Reviewer 1 Report

Comments and Suggestions for Authors

The authors described a vaccine approach to limit atherosclerotic progression in mice using peptide(s) from the TRPM2, a cation channel protein implicated in vascular inflammation.  Based on sequence homology analyses, they selected a pig-origin peptide for most of the studies including active vaccination by formulating the peptide in the standard alum adjuvant, a well-known adjuvant approved for human use. The vaccine formulation and immunization protocols are standard as are the immune outcome measurements (e.g. ELISA). It is reported that the maximum efficacy, measured using the en face Oil Red O staining procedure, was a net 10% reduction/suppression of atherosclerotic progression.  It is not clear how significant is this reduction because it is not compared to any standards or even the LDL-peptide based vaccines studied/reported by others in the literature.  Therefore, it is difficult to agree the importance and novelty of this vaccine approach.  Further, it is not clear from what is presented in this manuscript whether there are any long-term benefits. Other data in terms of using the peptide to generate rabbit antibodies and testing for reactivity seem replication of their previous published work. Overall, the study despite having been performed using standard methodologies, lacks novelty and clear understanding of translatable long-term clinical benefits. 

Author Response

Reviewer’s Comments: The authors described a vaccine approach to limit atherosclerotic progression in mice using peptide(s) from the TRPM2, a cation channel protein implicated in vascular inflammation.  Based on sequence homology analyses, they selected a pig-origin peptide for most of the studies including active vaccination by formulating the peptide in the standard alum adjuvant, a well-known adjuvant approved for human use. The vaccine formulation and immunization protocols are standard as are the immune outcome measurements (e.g. ELISA). It is reported that the maximum efficacy, measured using the en face Oil Red O staining procedure, was a net 10% reduction/suppression of atherosclerotic progression.  It is not clear how significant is this reduction because it is not compared to any standards or even the LDL-peptide based vaccines studied/reported by others in the literature.  Therefore, it is difficult to agree the importance and novelty of this vaccine approach.  Further, it is not clear from what is presented in this manuscript whether there are any long-term benefits. Other data in terms of using the peptide to generate rabbit antibodies and testing for reactivity seem replication of their previous published work. Overall, the study despite having been performed using standard methodologies, lacks novelty and clear understanding of translatable long-term clinical benefits. 

Summary: Thank you so much for taking the time to review this manuscript. Below are our detailed responses, along with the corresponding revisions and corrections, which are highlighted or tracked in the re-submitted files.

Answer: First, we apologize that we have not explained well enough about the novelty of the present study. Let me try to rephrase in two short paragraphs below.

Briefly, vaccine-based active immunization is an exciting strategy for atherosclerosis treatment/prevention. But most of the previous atherosclerosis-related vaccines have been designed to target LDL-related antigens, such as oxLDL (Moreno-Gonzalez MA et al., Nano Today. 2023;50:101822; Habets KLL et al., Cardiovasc Res 2010;85:622–630), apolipoprotein B-100 (a major component of LDL proteins) (Chyu KY et al., JCI Insight. 2022;7(11):e149741; Kimura T et al.. Circulation. 2018;138(11):1130-1143; Chyu KY et al., Biochem Biophys Res Commun. 2005;338(4):1982-9), proprotein convertase subtilisin/kexin type-9 serine protease (PCSK9) (Wu D et al., Cardiovasc Drugs Ther. 2021;35(1):141-151). Only very recently, studies have begun to explore other vaccination targets such as ANGPTL3 (Fukami H et al., Cell Rep Med. 2021;2(11):100446) and COL6A6 (Tang D et al., Cells. 2024;13(18):1589). Note that although researchers have been exploring atherosclerosis-related vaccines for about two decades (Moreno-Gonzalez MA et al., Nano Today. 2023;50:101822), up to the present these atherosclerosis-related vaccines are still in preclinical stages using animal models. None of these vaccines has been approved for clinical usage. The idea of developing vaccination strategies to treat atherosclerotic cardiovascular disease is still in its infancy. In the present study, we designed a peptide vaccine against a new target TRPM2, a ROS-sensitive Ca2+ channel that is only very recently found to be critically important for atherosclerotic development. Our results showed that a pig-origin TRPM2 peptide with standard alum adjuvant can effectively suppress the atherosclerotic progression in a mouse model of atherosclerosis. Note that the mechanisms of TRPM2 involvement in atherosclerosis are very different from that of LDL proteins and PCSK9. TRPM2 mediates Ca2+ entry, thereafter facilitates ROS-mediated macrophage infiltration and vascular inflammation (Zhang Y et al., Cells 2022;11(9):1423; Zong P, et al., Nat Cardiovasc Res. 2022;1(4):344-360). In contrast, LDL proteins and PCSK9 participate in cholesterol uptake. Therefore, our current study provides a novel target (TRPM2) for peptide vaccine-based anti-atherosclerotic strategy. This vaccine targets different steps (Ca2+ entry, ROS-mediated vascular inflammation, etc.) of atherosclerotic progression compared to that of previous vaccines (mostly targeting LDL uptake). Therefore, we believe that the novelty of our current study is very clear.

As for the effectiveness of different vaccines in suppressing atherosclerotic lesion, the reports from different groups vary considerably, ranging from relative reduction in atherosclerotic lesion by ~15% to ~60% with most reports at ~30-40% (Nettersheim FS et al., Cells. 2020;9(12):2560; Chyu KY et al., JCI Insight. 2022;7(11):e149741; Kimura T, et al., Circulation. 2018;138(11):1130-1143). Note that most of these previous vaccines target against LDL-related antigens (Moreno-Gonzalez MA, et al, Nano Today. 2023;50:101822; Nilsson J, Atherosclerosis. 2021;335:89-97; Nettersheim FS et al., Cells. 2020;9(12):2560). For comparison, in our study, TRPM2-based peptide vaccine P1 reduced the atherosclerotic lesion by 26%, which is in similar range of previous LDL-targeting vaccines. Note also that researchers have been developing/improving LDL-targeting vaccines for two decades with hundreds of publications (Moreno-Gonzalez MA et al., Nano Today. 2023;50:101822; Nilsson J, Atherosclerosis. 2021;335:89-97; Nettersheim FS, et al., Cells. 2020;9(12):2560). Therefore, we believe that our current first attempt in developing TRPM2-based peptide with 26% effectiveness can already be considered very good. 

Most of these discussions have been added to into the manuscript, either in “Introduction Section (the third paragraph)” or in “Discussion Section (the first and the last paragraph)”.

Reviewer 2 Report

Comments and Suggestions for Authors

The manuscript entitled "Active immunization using TRPM2 peptide vaccine attenuates atherosclerotic progression in a mouse model of atherosclerosis" is well written in clear understandable English and covers an important theme: the searching of potential anti-atheroscletoric vaccine. In general, the manuscript is good performed, however I have few questions for the authors.

Major

1. It is interesting to see the efficacy of the vaccine for Apoe mice, however this model is quite artificial for humans. It requires knocjout mice and specific diet. It is not typical for humans. Moreover, the main goal in vaccine production is that the produced vaccine will be safe for humans (animals). I think additional controls are required: WT at normal diet vaccinated by at least one more effective vaccine. If there are some side effects?  

2. Chapter Statystical analysis is poor written. You should state in this chapter how many animals were in each group. Why did you used t-test? How did you checked that distribution is normal? 

Minor:

1. Line 113 distilled water?

2. Line 122. Why you decided to use celiac vein for blood collection? Why not to use tail vein?

3. Line 152 (and the same below in the manuscript) "0Ca". I suppose that it is better to write Ca2+-free

4. Lines 260-261. Why you suppose that mice died due to the immunization? What are the proofs of this?

5. Redraw fig 1a. It is difficult to understand from that which groups were used.

Author Response

General Comments: The manuscript entitled "Active immunization using TRPM2 peptide vaccine attenuates atherosclerotic progression in a mouse model of atherosclerosis" is well written in clear understandable English and covers an important theme: the searching of potential anti-atherosclerotic vaccine. In general, the manuscript is good performed, however I have few questions for the authors.

 Summary: Thank you so much for taking the time to review this manuscript. Below are our detailed responses, along with the corresponding revisions and corrections, which are highlighted or tracked in the re-submitted files.

Major point #1:

It is interesting to see the efficacy of the vaccine for Apoe mice, however this model is quite artificial for humans. It requires knockout mice and specific diet. It is not typical for humans. Moreover, the main goal in vaccine production is that the produced vaccine will be safe for humans (animals). I think additional controls are required: WT at normal diet vaccinated by at least one more effective vaccine. If there are some side effects?  

Answer: Thank you for your comments. We fully agree that this atherosclerotic model of Apoe KO plus high cholesterol feeding is artificial and not typical for humans. However, this is also the most wild-used animal model of atherosclerosis. It mimics the human atherosclerotic situation in that both can be attributed to blood high LDL.

We also understand your request of additional control. However, it is well documented that WT mice with normal diet will not develop atherosclerotic lesion due to relative short life expectation of mice (Schreyer SA, et al., Atherosclerosis. 1998;136:17–24; Getz GS et al., Arterioscler Thromb Vasc Biol. 2006;26:242-9). That is the reason of why scientists developed ApoE KO mice in the first place to investigate atherosclerosis (Getz GS, et al., J Lipid Res. 2016;57(5):758-66). Furthermore, we did have a control of “Apoe KO mice at normal diet”, in which we did not see any atherosclerotic lesion in aortas (Figure 2a). In light of these background information, we believe that it may not be necessary to do another control of WT mice with normal diet. Please also consider is that the whole experimental cycle of atherosclerosis takes at least 7-8 months. We hope that you can be lenient on this point.

We appreciate your attention on the safety of the vaccine, especially after COVID-19 pandemic. During the stage of experimental design, we already considered the safety issue and chose the strategy of “peptide vaccine”, which are known to be highly safe compared to other types of vaccines (Li W et al., Vaccines (Basel). 2014;2(3):515-36). Peptide vaccines generally could not cause serious systemic adverse events. The most common systemic adverse events related to the peptide-based vaccines are erythema and induration at the injection site, which are easy to be reversed (Liu W, et al., Cell Prolif. 2021;54(5):e13025). Furthermore, at this stage we only used naked peptides with most popular KLH conjugation and a safe adjuvant aluminum hydroxide, all of which are not expected to have toxicity.

However, there are reports that an overactive immune system can sometimes trigger cytokine storm, causing excessive systemic inflammation and harmful damage (Ernzen K et al., Stem Cell Rev Rep. 2021;17(6):2107-2119). Therefore, we examined whether P1 and H2 peptide vaccines could elicit excessive inflammatory response. The results showed that P1 peptide vaccine had no effect on serum levels of inflammatory cytokines, including TNF-a, IL-1b and IL-6 (Supplemental Figure 2). However, H2 peptide vaccine caused certain increase in serum level of TNF-α, but not in serum levels of IL-1β and IL-6 (Supplemental Figure 2). Therefore, at least we did not observe harmful excessive inflammatory response to P1, which the most promising peptide vaccine for atherosclerosis treatment. 

Major point #2: Chapter Statistical analysis is poor written. You should state in this chapter how many animals were in each group. Why did you used t-test? How did you check that distribution is normal? 

Answer: We added more description/explanation in the chapter of Statistical analysis. Animal number is also added.

Normality test was performed by D'Agostino & Pearson omnibus normality test (Prism software recommended) or Shapiro-Wilk test or Kolmogorov-Smirnov test.

Minor Points:

  1. Line 113 distilled water?

Answer: Corrected. Thank you!

  1. Line 122. Why you decided to use celiac vein for blood collection? Why not to use tail vein?

Answer: Blood is taken for measurement of lipid profile and antibody titration. Measurement of blood lipid profile needs substantial amount of blood. We could only collect ~50 µl blood from tail vein, which was not enough for the measurement. Therefore, for blood lipid profile measurement, the blood was taken for celiac vein at the experimental end point when mice were sacrificed.

For measurement of blood antibody titration, we used Elisa kit, which needs less amount of blood. Therefore, here we draw blood from orbital sinus, where we could collect up to 300 µl blood per mouse.

  1. Line 152 (and the same below in the manuscript) "0Ca". I suppose that it is better to write Ca2+-free

Answer: Corrected.

  1. Lines 260-261. Why you suppose that mice died due to the immunization? What are the proofs of this?

Answer: This was only our speculation without any proof. As explained previously, peptide vaccines are known to be safe and generally could not cause serious systemic adverse events. The most common systemic adverse events related to the peptide-based vaccines are erythema and induration at the injection site, which are easy to be reversed (Liu W et al., Cell Prolif. 2021;54(5):e13025).

Vaccines are intended to induce immune responses including cellular immunity and humoral immunity. However, there are reports that an overactive immune system can sometimes trigger cytokine storm, causing excessive systemic inflammation and tissue damage (Ernzen K et al., Stem Cell Rev Rep. 2021;17(6):2107-2119). Therefore, we examined whether H2 peptide vaccine could elicit excessive production of inflammatory cytokines including TNF-α, IL-1β and IL-6. The results showed that H2 peptide vaccines caused certain increase in serum level of TNF-α, but not in serum levels of IL-1β and IL-6 (Supplemental Figure 2). Because the effect of H2 peptide vaccine on the production of inflammatory cytokines was not very consistent, it is unlikely that excessive immune responses was the main reason for the animal death. Further studies are needed to find the underlying reason. These texts are added to the relevant Result Section.

  1. Redraw fig 1a. It is difficult to understand from that which groups were used.

Answer: Corrected.

Reviewer 3 Report

Comments and Suggestions for Authors

Fan Ying et al. submitted an interesting work about TRPM2 peptide vaccine. The topic was less frequently discussed in the field, yet of a certain significance. The submission fell within the scope of Vaccines. The reviewer suggested a Major Revision for this paper. Detailed comments:

1.       The full name of TRPM2 should be provided.

2.       A scheme illustrating the whole picture of this study should be added at the end of the Introduction Section.

3.       How did the authors ensure the structure of the designed peptide? Any characterizations like MS?

4.       The stability of the designed peptide prior to administration should be determined.

5.       Scale bars of micrographs in Figure 4 must be supplemented.

6.       Discussion upon clinical translation and industrialization aspects should be added.

7.       Please consider to cite recent papers published in 2024 and 2025.

Round 2

Reviewer 1 Report

Comments and Suggestions for Authors

The authors provided additional literature details to support their selection of the vaccine candidate(s). The current study is preliminary in terms of moderate protection, mainly from one of the different peptides tested (P1), while other peptides like the H2 caused concerns related to cytokine storm.  Thus, it is unclear whether the vaccine promoted by the authors in this manuscript refers to which peptide for further development.  Therefore, it is recommended that the authors make an effort to further describe these limitations in the discussion. The very last sentence in the conclusions section at the end of the manuscript text (lines 456-457) should be removed for reasons of English language and ambiguous. 

Reviewer 2 Report

Comments and Suggestions for Authors

Can be accepted.

Author Response

Reviewer,s comments: can be accepted. So there is no need for response.

Reviewer 3 Report

Comments and Suggestions for Authors

Thanks for your revision.

Author Response

Reviewer,s comments: thanks for revision. So there is no need for response.